# Referring Transformer: A One-step Approach to Multi-task Visual Grounding

**Muchen Li**[1,2]
muchenli@cs.ubc.ca

**Leonid Sigal**[1,2,3,4]
lsigal@cs.ubc.ca

[1]Department of Computer Science, University of British Columbia
[2]Vector Institute for AI      [3]CIFAR AI Chair      [4]NSERC CRC Chair

## Abstract

As an important step towards visual reasoning, visual grounding (*e.g.*, phrase localization, referring expression comprehension / segmentation) has been widely explored. Previous approaches to referring expression comprehension (REC) or segmentation (RES) either suffer from limited performance, due to a two-stage setup, or require the designing of complex task-specific one-stage architectures. In this paper, we propose a simple one-stage multi-task framework for visual grounding tasks. Specifically, we leverage a transformer architecture, where two modalities are fused in a visual-lingual encoder. In the decoder, the model learns to generate contextualized lingual queries which are then decoded and used to directly regress the bounding box and produce a segmentation mask for the corresponding referred regions. With this simple but highly contextualized model, we outperform state-of-the-art methods by a large margin on both REC and RES tasks. We also show that a simple pre-training schedule (on an external dataset) further improves the performance. Extensive experiments and ablations illustrate that our model benefits greatly from contextualized information and multi-task training.

## 1   Introduction

Multi-modal *grounding*[1] tasks (*e.g.*, phrase localization [1, 3, 9, 41, 48], referring expression comprehension [17, 19, 24, 26, 29, 30, 37, 51, 52, 55, 56] and segmentation [6, 18, 20, 21, 29, 38, 53, 56]) aim to generalize traditional object detection and segmentation to localization of regions (rectangular or at a pixel level) in images that correspond to free-form linguistic expressions. These tasks have emerged as core problems in vision and ML due to the breadth of applications that can make use of such techniques, spanning image captioning, visual question answering, visual reasoning and others.

The majority of multi-modal grounding architectures, to date, take the form of two-stage approaches, inspired by Faster RCNN [44] and others, which first generate a set of image region proposals and then associate/ground one, or more, of these regions to a phrase by considering how well the content matches the query phrase. Context among the regions and multiple query phrases, which often come parsed from a single sentence, has also been considered in various ways (*e.g.*, using LSTM stacks [9], graph neural networks [1] and others). More recent variants leverage pre-trained multi-modal Transformers (*e.g.*, ViLBERT [33, 34]) to fine-tune to the grounding tasks. Such models have an added benefit of being able to learn sophisticated cross-modal feature representations from external

---

[1]Grounding and referring expression comprehension have been used interchangeably in the literature. While the two terms are indeed trying to characterize the same task of associating a lingual phrase with an image region, there is a subtle difference in that referring expressions tend to be unique and hence need to be grounded to a single region, *e.g.*, "*person in a red coat next to a bus*". The grounding task, as originally defined in [41], is more general where a lingual phrase may be ambiguous and therefore grounded to multiple regions, *e.g.*, "*a person*".

35th Conference on Neural Information Processing Systems (NeurIPS 2021).

large-scale data, which further improve the performance. However, a significant limitation of all such two-stage methods is their inability to condition the proposal mechanism on the query phrase itself, which inherently limits the upper bound of performance (see Table 3 in [51]).

To address these limitations, more recently, a number of one-stage approaches have been introduced [22, 36, 51, 52]. Most of these take inspiration from Yolo [43] and the variants, and rely on more integrated visual-linguistic fusion and a dense anchoring mechanism to directly predict the grounding regions. While this alleviates the need for a proposal stage, it instead requires somewhat ad hoc anchor definitions, often obtained by clustering of labeled regions, and also limits ability to contextualize grounding decisions as each query phrase is effectively processed independently. Finally, little attention in the literature has been given to leveraging relationship among the REC and RES tasks.

In this work we propose an end-to-end one-stage architecture, inspired by the recent DETR [2] detection framework, which is capable of simultaneous language grounding at both a bounding-box and segmentation level, without requiring dense anchor definitions. This model also enables contextualized reasoning by taking into account the entire image, all referring query phrases of interest and (optionally) lingual context (*e.g.*, a sentence from which referring phrases are parsed). Specifically, we leverage a transformer architecture, with a visual-lingual encoder, to encode image and lingual context, and a two-headed (detection and segmentation) custom contextualized transformer decoder. The contextualized decoder takes as input learned contextualized phrase queries and decodes them directly to bounding boxes and segmentation masks. Implicit 1-to-1 correspondence between input referring phrases and resulting outputs also enables a more direct formulation of the loss without requiring Hungarian matching. With this simple model we outperform state-of-the-art methods by a large margin on both REC and RES tasks. We also show that a simple pre-training schedule (on an external dataset) further improves the performance. Extensive experiments and ablations illustrate that our model benefit greatly from the contextualized information and the multi-task training.

**Contributions.** Our contributions are: (1) We propose a simple and general one-stage transformer-based architecture for referring expression comprehension and segmentation. The core of this model is the novel transformer decoder that leverages contextualized phrase queries and is able to directly decode those, subject to contextualized image embeddings, into corresponding image regions and segments; (2) Our approach is unique in enabling simultaneous REC and RES using a single trained model (the only other method capable of this is [36]); showing that such multi-task learning leads to improvements on both tasks; (3) As with other transformer-based architectures, we show that pre-training can further improve the performance and both vanila and pre-trained models outperform state-of-the-art on both tasks by significant margins (up to 8.5% on RefCOCO dataset for REC and 19.4% for RES). We also thoroughly validate our design in detailed ablations.

## 2   Related works

**Referring Expression Comprehension (REC).**   REC focuses on producing an image bounding box tightly encompassing a language query. Previous two-staged works [17, 19, 29, 30, 56] reformulate this as a ranking task with a set of candidate regions predicted from a pre-trained proposal mechanism. Despite achieving great success, the performance of two-staged methods is capped by the speed and accuracy of region proposals in the first stage. More recently, one-stage approaches [26, 51, 52] have been used to alleviate the aforementioned limitations. Yang *et al.* [51, 52] proposed to fuse query information with visual features and pick the bounding box with maximum activation scores from YOLOv3 [43]. Yang *et al.* [50] explore language structure guided propagation in the context of one stage grounding. Liao *et al.* [26] utilizes CenterNet [11] to perform correlation filtering for region center localization. However, such methods either require manually tuned anchor boxes or suffer from semantic loss due to modality misalignment. In contrast, our model learns to better align modalities using a cross-modal transformer and directly decode bounding boxes for each query.

**Referring Expression Segmentation (RES).**   Similar to REC, RES, proposed in [18], aims to predict segmentation masks to better describe the shape of the referred region. A typical solution for referring expression segmentation is to fuse multi-modal information with a segmentation network (*e.g.*, [16, 31]) and train it to output the segmented masks [18, 29, 38, 53, 56]. More recent approaches focus on designing module to enable better multi-modal interactions, *e.g.*, progressive multi-scale fusion used in [21] and cross-modal attention block used in [20]. Since localization information matters in predicting instance segmentations (as noted in Mask RCNN [16]), very recent work [22] aims to explicitly localize object before doing segmentation. Despite the relatively high performance

being achieved in RES, existing approaches still struggle to determine the correct referent region and tend to output noisy segmentation results with an irregular shape, while our model is able to produce segmentations with fine-grained shapes even on challenging scenarios with occlusions or shadows.

**Multi-task Learning for REC and RES.** Multi-task learning is widely applied in object detection and segmentation [2, 16], often, by leveraging shared backbone and task-specific heads. Building on this idea, Luo *et al.* [36] proposed a multi-task collaborative network (MCN) to jointly address REC and RES. They introduce consistency energy maximization loss that constrains the feature activation map in REC and RES to be similar. While our model is also set up to learn REC and RES tasks jointly, we argue that an explicit constraint tends to downplay the quality of the final predicted mask since the feature map from the REC branch can blur out fine-grained region shape information needed by the RES branch (see Figure 2). Hence, we use an implicit constraint where tasks head of REC and RES are trained to output corresponding bounding box and mask from the same joint multi-modal representation. We illustrate that our model can benefit from multi-task supervision, leading to more accurate results as compared to single-task variants.

**Pretrained Multi-modal Transformers.** Transformer-based pretrained models [5, 12, 33, 47, 54] have recently showed strong potential in multi-modal understanding. LXMERT [47] and ViLBERT [33] use two stream transformers with cross-attention transformer layers on top for multimodal fusion. More recent works, [5, 12] advocate a single-stream design to fuse two modalities earlier. The success of the aforementioned models can largely be attributed to the cross-modal representations obtained by multi-task pretraining on a large amount of aligned image-text pairs. Despite state-of-the-art performance of such models on the downstream REC task, these models, fundamentally, are still a form of a two-stage pipeline where image features are extracted using pretrained detectors or proposal mechanisms. We focus on a one-stage architecture variant that allows visual and lingual features to be aligned at the early stages. Although the focus of our work is not to design a better pretraining scheme, we show that our model can outperform the existing state-of-the-art with proper pretraining.

**Transformer-based Detectors.** More recently, DETR [2] and its variants [13, 58], were proposed to enable end-to-end object detection. DETR reformulates detection as a set prediction tasks and uses transformers to decode learnable queries to bounding boxes. Despite state-of-art performance, DETR is disadvantaged by its optimization difficulty and, usually, extra-long training time. While adopting a similar pipeline, our model focus on aligning different modalities to generate contextualized expression-specific referring queries. We also design our model to get rid of Hungarian matching loss by leveraging one-to-one correspondences between predicted bounding boxes and referring expressions, which leads to faster convergence for our model.

## 3 Approach

Given an image $\mathcal{I}$ and a set of query phrases $\mathcal{Q}_p = \{\mathbf{p}_i\}_{i=1,...,M}$, that we assume to come from an (optional) contextual text source[2] $\mathcal{Q}$, our goal is to predict a set of bounding boxes $\mathcal{B} = \{\mathbf{b}_i\}_{i=1,...,M}$ and corresponding segmentation masks $\mathcal{S} = \{\mathbf{s}_i\}_{i=1,...,M}$, one for each query phrase $i$ that localizes that phrase in the image. Note, $M$ is the number of phrases / referring expressions for a given image $\mathcal{I}$ and is typically between 1 and 16 for the Flickr30k [41] dataset.

As shown in Figure 1, our referring transformer is composed of four components. Given an image-(con)text pair, $< \mathcal{I}, \mathcal{Q} >$, a *cross-modal encoder* generates joint image-text embeddings for each visual and textual token – feature columns and word embeddings respectively. Query phrases $\mathcal{Q}_p$ and image-text embeddings are then fed into a *query encoder* which produces query phrase embeddings. The *decoder* jointly reasons across all these query phrase embeddings and decodes multi-task feature, which is then sent to the *detection* and *segmentation head* to produce a set of boxes $\mathcal{B}$ and masks $\mathcal{S}$. The result is a one-staged end-to-end model that solves the REC and RES tasks at the same time. We will now introduce constituent architectural components for the four stages briefly described above.

### 3.1 Feature Extraction

**Visual & Text Backbone.** Starting from an initial image $\mathcal{I} \in \mathbb{R}^{3 \times H_0 \times W_0}$, we adopt the widely used ResNet [15] to generate its low-resolution feature map $\mathbf{f}_I \in \mathbb{R}^{C_i \times HW}$. For the corresponding

---

[2]For example, a sentence from which noun phrases/referring expressions $\mathbf{p}_i$ were parsed. Where contextual text source is unavailable and only one phrase exists, we simply let $\mathcal{Q} = \mathcal{Q}_p$.

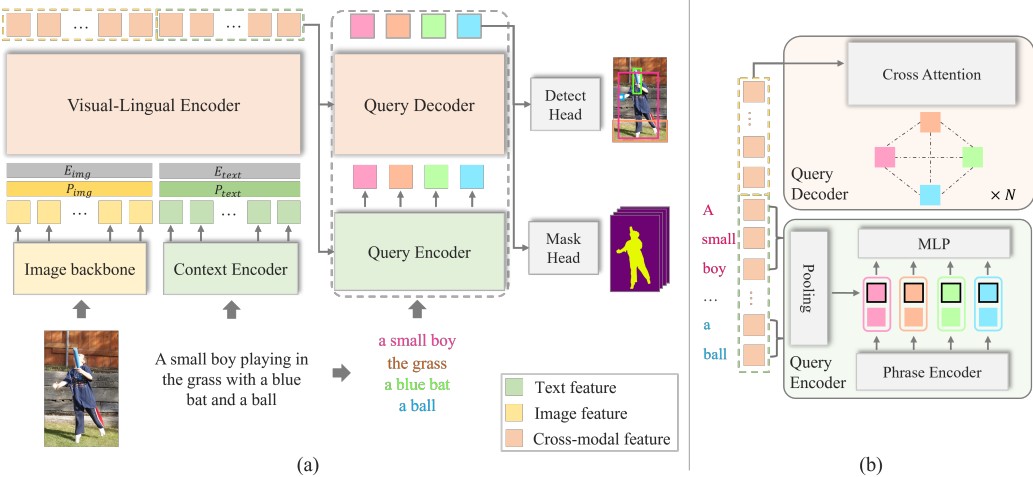

Figure 1: **Referring Transformer.** An overview of the proposed architecture is shown in (a). For an image and (con)text input, a visual-lingual encoder is used to refine image features, extracted from a convolutional backbone, and lingual features, extracted by a BERT. A query encoder and decoder produce features for REC and RES heads, given multi-modal features and query phrases. The detailed structure of the query encoder and decoder is shown in (b). Colored squares denote embeddings for corresponding query phrases.

expression or sentence, we use the uncased base of BERT [8] to obtain the representation $\mathbf{f}_Q \in \mathbb{R}^{C_t \times N}$, while $N$ is the length of the input context sentence.

**Visual-Lingual Encoder.** The visual-lingual encoder is designed to fuse information from multimodal sources. For cross-modality encoding, we use a transformer encoder based model, which is composed of 6 transformer encoder layers. Specifically, given both image and text features, multi-layer perceptrons are applied first to project different modalities to a joint embedding space with a hidden dimension of $C$. To provide transformer encoders with positional information, we follow [2, 8] to add cosine positional embedding $P_{img}$ for image features and learnable positional embedding $P_{text}$ for text features. We then concatenate the projected features into a single sequence $\mathbf{f} \in \mathbb{R}^{C \times (HW+N)}$. To distinguish between modalities, we also design a learnable modal label embedding $E_{label} : \{E_{img}, E_{text}\}$ which is added to the original sequences. The visual-lingual encoder then takes a sequence as input, and $\{P_{img}, P_{text}, E_{label}\}$ are fed into each encoder layer. The encoder output is a multi-modal feature sequence $\mathbf{f}_{vl} \in \mathbb{R}^{C \times (HW+N)}$.

## 3.2 Query Encoder and Decoder

The query encoder and decoder aims to encode / decode query phrases, conditioned on visual-lingual features from the encoder, into an output bounding box and segmentation. In this stage, we first generate embeddings corresponding to each query phrase. These query phrase embeddings are then fed into the decoder together with the visual-lingual features from the encoder to generate outputs.

**Encoding Query Phrases.** To enable the decoder to generate the desired output (bounding box and/or segmentation) the queries must encode several bits of crucial information. Mainly, (1) encoding of the query phrases, (2) image encoding and (3) phrase-specific optional (con)text information. For phrase encoding in (1) we use a BERT model with pooling heads which share weights with (con)text encoder; this results in the phrase feature vector $\mathbf{f}_{\mathbf{p}_i} \in \mathbb{R}^C$ for the $i$-th referring phrase. We note that because we use visual-lingual encoder, (2) and (3) are jointly encoded in multi-modal features $\mathbf{f}_{vl}$ described in Section 3.1 above. However, $\mathbf{f}_{vl}$ is phrase-agnostic encoding of the image and (con)text. To generate phrase-specific context, given a phrase $\mathbf{p}_i$, average pooling is used to extract the phrase-specific context information $\mathbf{f}_c(\mathbf{p}_i)$ from the visual-lingual feature sequence as follows:

$$\mathbf{f}_c(\mathbf{p}_i) = \frac{\sum \mathbf{f}_{vl}[l_{\mathbf{p}_i} : r_{\mathbf{p}_i}]}{r_{\mathbf{p}_i} - l_{\mathbf{p}_i}} \tag{1}$$

where $l_{\mathbf{p}_i}$ and $r_{\mathbf{p}_i}$ denotes the left and right bounds of phrase $\mathbf{p}_i$ in the original (con)text sentence. Finally, given phrase encoding $\mathbf{f}_{\mathbf{p}_i}$ and phrase-specific context $\mathbf{f}_c(\mathbf{p}_i)$ we construct our phrase queries

using a multi-later perceptron:

$$\widehat{Q_{\mathbf{p}_i}} = \mathtt{MLP}\left([\mathbf{f}_c(\mathbf{p}_i); \mathbf{f}_{\mathbf{p}_i}]\right) + E_p, \tag{2}$$

where $E_p \in \mathbb{R}^C$ is a learnable embedding which serves as a bias to the formed query.

**Decoding.** In the decoder, self attention layers are used to enable information flow in a dense connected graph of phrase queries. This allows phrase queries to contextualize and refine each other; the inspiration for this step is taken from [1]. After that, a cross attention layer decodes visual-lingual information given the updated phrase query and feature sequence from the encoder. The design of our decoder is similar to the transformer decoder, except attention is non-causal.

### 3.3 Multi-task training

In this section, we demonstrate how the decoded phrase-specific query features can be naturally used to train multiple heads for different referring tasks (regression for REC and segmentation for RES).

**Referring Comprehension/Detection (REC).** For referring detection tasks, the final output is computed by a simple two-layer perceptron over the decoded phrase-specific query features. We let the detection head directly output center coordinates $\tilde{\mathbf{b}} = (x, y, h, w)$ for the referred image. To supervise the training, we use a weighted sum of an L1 loss and a Generalized IOU loss [45]:

$$\mathcal{L}_{det} = \lambda_{iou}\mathcal{L}_{iou}(\mathbf{b}, \tilde{\mathbf{b}}) + \lambda_{L1}||\mathbf{b} - \tilde{\mathbf{b}}||_1. \tag{3}$$

The $\lambda_{iou}$ and $\lambda_{L1}$ control the relative weighting of the two losses in the REC objective.

**Referring Segmentation (RES).** Following previous work [2], we design an FPN-like architecture to predict a referring segmentation mask for each phrase expression. Attention masks from the decoder and image features from the visual-lingual encoder are concatenated as the FPN input, while features from different stages of image backbones are used as skip connections to refine the final output. The last linear layer project the upsampled feature to a single channel heatmap and a sigmoid function is used to map the feature to mask scores $\tilde{\mathbf{s}} \in \mathbb{R}^{H_0/4 \times W_0/4}$. The loss for training RES task is:

$$\mathcal{L}_{seg} = \lambda_{focal}\mathcal{L}_{focal}(\mathbf{s}, \tilde{\mathbf{s}}) + \lambda_{dice}\mathcal{L}_{dice}(\mathbf{s}, \tilde{\mathbf{s}}). \tag{4}$$

Here $\mathcal{L}_{focal}$ is the focal loss for classifying pixels used in [28], $\mathcal{L}_{dice}$ is the DICE/F-1 loss proposed in [39]; $\lambda_{focal}$ and $\lambda_{dice}$ are hyper-parameters controlling the relative importance of the two losses.

**Joint Training.** While it is possible to train referring segmentation and referring detection tasks separately, we find that joint training is highly beneficial. Therefore the combined training loss which we optimize is $\mathcal{L} = \mathcal{L}_{seg} + \mathcal{L}_{det}$.

**Pretraining the Transformer.** Transformers are generally data hungry and requires a lot of data to train [5, 33]. Although in this paper we do not use a large pretraining model and a lot of data. We found that simple pretraining strategy on the region description splits of Visual-Genome dataset [25] makes our model achieve comparable and even better performance against some of state-of-the-art pretrained models. Interestingly, we found that although there is no ground truth segmentation provided in Visual-Genome, the RES task can still benefit greatly from pretrained models, likely due to the fine-tuned multi-task representation.

## 4 Experiments

### 4.1 Datasets

**RefCOCO/RefCOCO+/RefCOCOg (REC&RES).** RefCOCO, RefCOCO+ [55] and RefCOCOg [40] are collections of images and referred objects from MSCOCO [27]. On RefCOCO and Ref-COCO+ we follow the split used in [55] and report scores on the validation, testA and testB splits. On RefCOCOg, we use the RefCOCO-umd splits proposed in [40].

**Flickr30k Entities (REC).** Flickr30k Entities [41] contains 31,783 images and 158k caption sentences with 427k annotated phrase. We use splits from [41, 42]. Bounding boxes and phrase annotations are consistent with the previous one-stage approaches [51, 52] for fair comparisons.

**ReferIt (REC).** The ReferItGame dataset [24] contains 20,000 images. We follow setup in [3] for splitting train, validation and test set; resulting in 54k, 6k and 6k referring expressions respectively.

Table 1: **Comparison on REC task.** Performance on RefCOCO/RefCOCO+/RefCOCOg datasets [55] is reported. Ours* denotes that pretraining is used. RN50 and RN101 refer to ResNet50 and ResNet101 [15] respectively; DN53 refers to DarkNet53 [43] backbone.

| Models | Visual Features | Pretrain Images | Multi-task | RefCOCO | | | RefCOCO+ | | | RefCOCOg | |
|---|---|---|---|---|---|---|---|---|---|---|---|
| | | | | val | testA | testB | val | testA | testB | val-u | test-u |
| *Two-stage:* | | | | | | | | | | | |
| CMN [19] | VGG16 | None | × | - | 71.03 | 65.77 | - | 54.32 | 47.76 | - | - |
| RvG-Tree [17] | RN101 | None | × | 75.06 | 78.61 | 69.85 | 63.51 | 67.45 | 56.66 | 66.95 | 66.51 |
| CM-Att-Erase [30] | RN101 | None | × | 78.35 | 83.14 | 71.32 | 68.09 | 73.65 | 58.03 | 67.99 | 68.67 |
| MAttNet [56] | RN101 | None | ✓ | 76.65 | 81.14 | 69.99 | 65.33 | 71.62 | 56.02 | 66.58 | 67.27 |
| NMTree [29] | RN101 | None | ✓ | 76.41 | 81.21 | 70.09 | 66.46 | 72.02 | 57.52 | 65.87 | 66.44 |
| *One-stage:* | | | | | | | | | | | |
| RCCF [26] | DLA34 | None | × | - | 81.06 | 71.85 | - | 70.35 | 56.32 | - | 65.73 |
| SSG [4] | DN53 | None | × | - | 76.51 | 67.50 | - | 62.14 | 49.27 | 58.80 | - |
| FAOA [51] | DN53 | None | × | 72.54 | 74.35 | 68.50 | 56.81 | 60.23 | 49.60 | 61.33 | 60.36 |
| ReSC-Large [52] | DN53 | None | × | 77.63 | 80.45 | 72.30 | 63.59 | 68.36 | 56.81 | 67.30 | 67.20 |
| MCN [36] | DN53 | None | ✓ | 80.08 | 82.29 | 74.98 | 67.16 | 72.86 | 57.31 | 66.46 | 66.01 |
| Ours | RN50 | None | ✓ | 81.82 | 85.33 | 76.31 | 71.13 | 75.58 | 61.91 | 69.32 | 69.10 |
| Ours | RN101 | None | ✓ | **82.23** | **85.59** | **76.57** | **71.58** | **75.96** | **62.16** | **69.41** | **69.40** |
| *Pretrained:* | | | | | | | | | | | |
| VilBERT[33] | RN101 | 3.3M | × | - | - | - | 72.34 | 78.52 | 62.61 | - | - |
| ERNIE-ViL_L[54] | RN101 | 4.3M | × | - | - | - | 75.89 | 82.37 | 66.91 | - | - |
| UNTIER_L[5] | RN101 | 4.6M | × | 81.41 | 87.04 | 74.17 | 75.90 | 81.45 | 66.70 | 74.86 | 75.77 |
| VILLA_L[12] | RN101 | 4.6M | × | 82.39 | 87.48 | 74.84 | 76.17 | 81.54 | 66.84 | 76.18 | 76.71 |
| Ours* | RN50 | 100k | ✓ | 85.43 | 87.48 | 79.86 | 76.40 | 81.35 | 66.59 | 78.43 | 77.86 |
| Ours* | RN101 | 100k | ✓ | **85.65** | **88.73** | **81.16** | **77.55** | **82.26** | **68.99** | **79.25** | **80.01** |

## 4.2 Implementing Details

We train our model with AdamW [32]. The initial learning rate is set to 1e-4 while the learning rate of image backbone and context encoder is set to 1e-5. We initialized weights in the transformer encoder and decoder with Xavier initialization [14]. For image backbone, we experiment with the popular ResNet-50 and ResNet-101 networks [15] where weights are initialized from corresponding ImageNet-pretrained models. For the context encoder and phrase encoder, we use an uncased version of BERT model [8] with weights initialized from pretrained checkpoints provided by HuggingFace [49]. For data augmentation, we scale images such that the longest side is 640 pixels and follow [51] to do random intensity saturation and affine transforms. We remove the random horizontal flip augmentation used in previous work [51] since we notice it causes semantic ambiguity on RefCOCO, likely due to relative location (*e.g.*, left of/right of) specific queries in the dataset.

On Flickr30k dataset, we set the maxium length of context sentence to 90 and maximum number of referring phrases to 16. On the ReferIt and the RefCOCO dataset, only phrase expressions are provided and the task aims to predict a single bounding box for each of the expressions. In those cases, the context sentence is taken as the referring phrase expression itself. We set the maximum length of context sentence on these two datasets to 40. To fairly compare with pretrained methods, we use region description split in the VisualGenome [25] to pretrain our model. The dataset contains approximately 100k images and we remove the images that appear in Flickr30k Entities and RefCOCO/RefCOCOg/RefCOCO+'s validation and test set to avoid potential test data leak. For all the pretrained methods, we train the model on pretraining dataset for 6 epoches. We find that longer pretraining schedule gives better performance, but since the focus of this paper is not on pretraining methods, we stick to shorter pretraining schedules to save computational resources. All experiments are conducted using 4 Nvidia 2080TI GPU with batch size as 32. For all the results given, we run experiments several times with random seeds and the error bars are within $\pm 0.5\%$.

## 4.3 Quantitative Analysis

**Evaluation Metrics.** For referring expression comprehension (REC), consistent with prior works, we use precision as the evaluation metric. We mark a referring detection as correct when the intersection-over-union (IoU) between the predicted bounding box and ground truth is larger than 0.5. For referring expression segmentation (RES), we reported the Mean IoU (MIoU) between the predicted segmentation mask and ground truth mask.

**REC and RES on RefCOCO/RefCOCO+/RefCOCOg.** Our model addresses REC and RES tasks jointly. We compare their respective performances with the state-of-the-art in Table 1 and 2.

Table 2: **Comparison on RES tasks.** Performance on RefCOCO/RefCOCO+/RefCOCOg datasets [55] is reported. Ours* denotes that pretraining is used. RN50 abd RN101 refer to ResNet50 and ResNet101 [15] respectively; DN53 refers to DarkNet53 [43] backbone.

| Methods | Backbone | RefCOCO | | | RefCOCO+ | | | RefCOCOg | | Inference |
|---|---|---|---|---|---|---|---|---|---|---|
| | | val | testA | testB | val | testA | testB | val | test | time(ms) |
| DMN [38] | RN101 | 49.78 | 54.83 | 45.13 | 38.88 | 44.22 | 32.29 | - | - | - |
| MAttNet [56] | RN101 | 56.51 | 62.37 | 51.70 | 46.67 | 52.39 | 40.08 | 47.64 | 48.61 | 378 |
| NMTree [29] | RN101 | 56.59 | 63.02 | 52.06 | 47.40 | 53.01 | 41.56 | 46.59 | 47.88 | - |
| Lang2seg [6] | RN101 | 58.90 | 61.77 | 53.81 | - | - | - | 46.37 | 46.95 | - |
| BCAM [20] | RN101 | 61.35 | 63.37 | 59.57 | 48.57 | 52.87 | 42.13 | - | - | - |
| CMPC [21] | RN101 | 61.36 | 64.53 | 59.64 | 49.56 | 53.44 | 43.23 | - | - | - |
| MCN+ASNLS [36] | DN53 | 62.44 | 64.20 | 59.71 | 50.62 | 54.99 | 44.69 | 49.22 | 49.40 | 56 |
| CGAN [35] | DN53 | 64.86 | 68.04 | 62.07 | 51.03 | 55.51 | 44.06 | 51.01 | 51.69 | - |
| LTS [22] | DN53 | 65.43 | 67.76 | 63.08 | 54.21 | 58.32 | 48.02 | 54.40 | 54.25 | - |
| Ours | RN50 | 69.94 | 72.80 | 66.13 | 60.9 | 65.20 | 53.45 | 57.69 | 58.37 | **38** |
| Ours | RN101 | 70.56 | 73.49 | 66.57 | 61.08 | 64.69 | 52.73 | 58.73 | 58.51 | _41_ |
| Ours* | RN50 | _73.61_ | _75.22_ | _69.80_ | _65.30_ | _69.69_ | _56.98_ | _65.70_ | _65.41_ | **38** |
| Ours* | RN101 | **74.34** | **76.77** | **70.87** | **66.75** | **70.58** | **59.40** | **66.63** | **67.39** | _41_ |

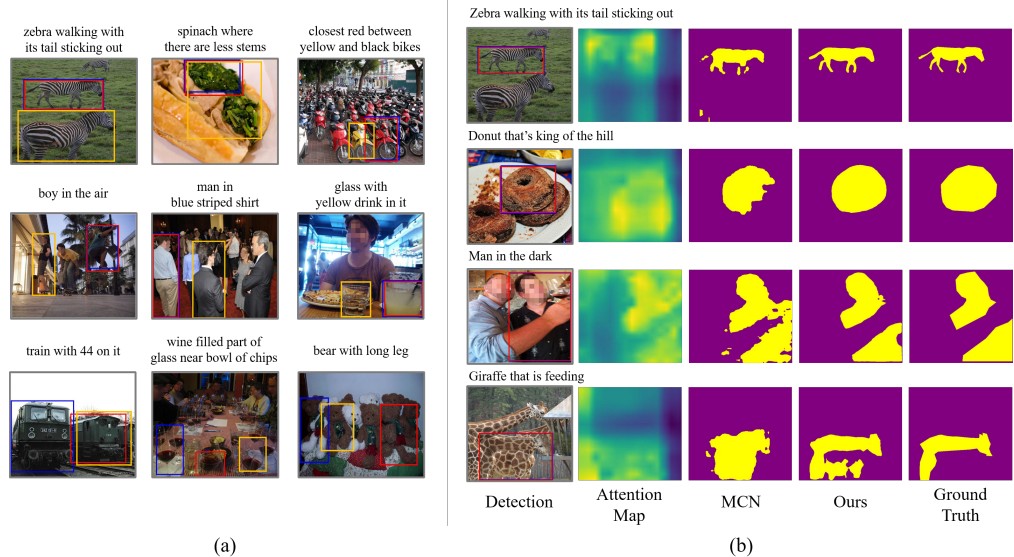

(a)                                                                (b)

Figure 2: **Qualitative Evaluation.** In (a) comparison to MCN [36] on REC is shown; orange, blue and red bounding boxes correspond to outputs from MCN, our model and the ground truth. In (b) similar comparison on RES is made. The attention map is drawn from the last layer of the decoder. We add mosaic to all human face to protect personal information.

In Table 1, the model is first compared with previous one-stage and two-stage approaches for REC. Without bells and whistles, we observe a consistent performance boost of $+2.7\%/+4\%/+2.1\%$ on RefCOCO, $+6.6\%/+4.3\%/+8.5\%$ on RefCOCO+ and $+4.4\%/+5.1\%$ on RefCOCOg. To compare with pretrained BERT methods, we use the pretraining strategy discussed in Section 3.3. As results show, our model achieves comprehensive advantage and shows distinct improvement on some splits, even compared to advance BERT models that use $40\times$ more data in pretraining. Table 2 illustrates results on RES task in terms of MIoU score. It can be seen that our model achieves the best performance; substantially better than the state-of-art. We further observe that pretraining on the REC task gives a huge performance boost to the RES task, even when no segmentation mask is used in pre-training. Multi-task training enables the model to leverage performance boost in one task to improve the other.

We show the inference time for our model in Table 2. Our model can directly decode all query phrases in an image in parallel, allowing it to reach real-time performance. Importantly, note the corresponding scores for our model in Table 1 and Table 2 are based on the output of a single multi-task model that predicts referring detection box and segmentation mask simultaneously. The only related work that shares this property is the MCN [36], which has substantially inferior performance.

Table 3: **Comparison with State-of-The-Art Methods.** Table illustrates performance on the test set of ReferItGame [24] and Flickr30K Entities [41] datasets in terms of top-1 accuracy (%).

| Models | Backbone | ReferItGame test | Flickr30K test | Inference time on Flickr30k(ms) |
|---|---|---|---|---|
| *Two-stage* | | | | |
| MAttNet [56] | RN101 | 29.04 | - | 320 |
| Similarity Net [48] | RN101 | 34.54 | 60.89 | 184 |
| CITE [42] | RN101 | 35.07 | 61.33 | 196 |
| DDPN [57] | RN101 | 63.00 | 73.30 | - |
| *One-stage* | | | | |
| SSG [4] | DN53 | 54.24 | - | 25 |
| ZSGNet [46] | RN50 | 58.63 | 58.63 | - |
| FAOA [51] | DN53 | 60.67 | 68.71 | 23 |
| RCCF [26] | DLA34 | 63.79 | - | 25 |
| ReSC-Large [52] | DN53 | 64.60 | 69.28 | 36 |
| Ours | RN50 | 70.81 | 78.13 | 37(14) |
| Ours | RN101 | 71.42 | 78.66 | 40(15) |
| Ours* | RN50 | 75.49 | 79.46 | 37(14) |
| Ours* | RN101 | **76.18** | **81.18** | 40(15) |

**REC on Flickr30k-Entities.**    For Flickr30k dataset, previous one-stage works [51, 52] extract short phrases from sentences and treated them as separate referring expression comprehension tasks. We argue that in this setting, queries are mostly short phrases and therefore cannot well reflect the model's ability to comprehend them in context. In contrast, our model, given an image and a caption (con)text sentence, aims to predict bounding boxes for *all* referred entities in the sentence. Doing so gives several advantages: 1. We are able to contextualize referring expressions given all other referring expressions and (con)text provided by the sentence. 2. Locations for all phrases can be inferred in one forward pass of the network, which saves a lot of computation as compared to previous one-stage approaches [51, 52] that process one phrase at a time. Note that our task formulation is consistent with some two-stage models [1, 9], but is unique for a one stage approach.

In Table 3 we compare our results with state-of-the-art methods. Without pretraining, we obtain a huge performance boost compared to both previous one-stage (+13.54%) and two-stage (+7.31%) state-of-the art methods. By using pretrained models, we observe that our model tends to generalize better on the test set and gives even better performance. We also provide comparison of inference time both per image and (per-expression), since our model can amortize inference across expressions. Per-expression, our inference time is substantially lower than all prior methods.

**REC on ReferIt.**    Since ReferIt is a relative small dataset, we use a slightly smaller model which contains 3 cross attention layers in the query decoder. The results are shown in Table 3. Our model is able to perform better, by a large margin, than even the latest one-stage methods. We also observe a consistent boost brought by pretraining.

## 4.4    Qualitative Analysis

In Figure 2, we show our qualitative comparison with previous state-of-the-art multi-task model – MCN [36]. The first two rows of Figure 2(a) show failure cases of MCN that can be better handled by our model. We observe that MCN appears to fail because it neglects some attributes in referring expression (*e.g.*, "*yellow* drink" and "*blue* striped shirt"), while our model is able to better model the query and pay attention to object attributes. In the last row, we shows several failure cases of our model. For the first case, the query requires the model to have the ability to recognize number "44". For the second and third case, there is visual ambiguity to identify the nearest glass to the bowl or to determine which bear(brown or white) has the longest leg.

In Figure 2(b), we show qualitative comparison in terms of referred segmentation mask. Compared to MCN, our model is able to output more detailed object shape and finer outlines. Moreover, our model shows the ability to handle shadows (*e.g.*, the right bottom of the donut) and occlusions (*e.g.*, the man occluded by another man's arm) and predict smoother segmentation mask. We also give a result on a challenging case in the last row, where the texture boundary of the two giraffe is hard to distinguish. Despite imperfections, our model is still able to focus on the giraffe's head in the foreground and performs much better than MCN. Part of our model's ability to generate fine-grained mask can be explained by better localization ability brought by the REC task, in which case the RES head can focus on tuning the shape and boundary of the mask.

Table 4: **Ablation studies.** Table on the top ablates our multi-task and pretraining schehme on RefCOCO/+/g validation set. Table on the bottom ablates on core components of our model.

| REC | RES | Pretrain | REC Acc↑ | RES Miou↑ | IE↓ |
|---|---|---|---|---|---|
|  | ✓ |  | - | 66.03 / 58.39 / 54.61 | 23.52% |
| ✓ |  |  | 81.08 / 70.02 / 68.15 | - | |
| ✓ | ✓ |  | **81.82 / 71.13 / 69.32** | **69.94 / 60.90 / 57.69** | **4.73%** |
| ✓ |  | ✓ | 85.01 / 75.46 / 77.96 | - | |
| ✓ | ✓ | ✓ | **85.43 / 76.40 / 78.43** | **73.61 / 65.30 / 65.70** | **4.48%** |

|  | Flickr30k |
|---|---|
| Model component | |
| -w/o *Query Decoder* | 49.38 |
| -w/o *Context Encoder* | 73.68 |
| Query Encoder | |
| -w/o *Context & Phrase Feature* | 42.05 |
| -w/o *Context Feature* | 76.64 |
| -w/o *Phrase Feature* | 77.02 |
| *Full model* | 78.13 |

Table 5: **Results on RefCOCO+ Dataset with Different Input Resolutions.** Our methods correspond to the RN50 model without pretraining.

| Models | Resolution | REC(prec@0.5) | | | RES(MIoU) | | |
|---|---|---|---|---|---|---|---|
| | | val | testA | testB | val | testA | testB |
| FAOA [51] | 256× 256 | 56.81 | 60.23 | 49.60 | - | - | - |
| ReSC-Large [52] | 256× 256 | 63.59 | 68.36 | 56.81 | - | - | - |
| CMPC [21] | 320× 320 | - | - | - | 49.56 | 53.44 | 43.23 |
| LTS [22] | 416× 416 | - | - | - | 54.21 | 58.32 | 48.02 |
| MCN [36] | 416× 416 | 67.16 | 72.86 | 57.31 | 50.62 | 54.99 | 44.69 |
| Ours | 256× 256 | 70.05 | 73.29 | 61.48 | 58.26 | 61.09 | 52.20 |
| Ours | 320× 320 | 70.03 | 73.23 | 61.52 | 58.42 | 61.48 | 52.34 |
| Ours | 416× 416 | 71.50 | 75.87 | 61.71 | 61.00 | 64.48 | 52.44 |
| Ours | 640× 640 | 71.58 | 75.96 | 62.16 | 61.08 | 64.69 | 52.73 |

## 4.5 Ablation Studies

We first consider the importance of the multi-task setup in Table 4 (top). Results indicate that multi-task training consistently boosts both REC and RES performance on Refcoco/+/g datasets by a considerable margin. More specifically, we observe that REC loss helps the transformer to better locate the referred object and converge faster in early stages of training. At the same time, RES loss aids the model with more fine-grained information on the shape of the referred region, which helps to further enhance the accuracy. IE here is *Inconsistency Error* metric originally used in [36] to measure the prediction conflict between the REC and RES task. We can see that joint training of RES and REC greatly reduce the inconsistency between the two tasks. Note that our model also has a much lower multi-task inconsistency compared to MCN [36], with a corresponding IE score of 7.54%(-40%). This shows that our model can do better collaborative learning.

Next, we validate the design of our network architecture. We report our scores on the Flickr30k test sets. In Table 4 (bottom), we ablate the model's major components and features used to form the query. Without (w/o) context encoder indicates that we directly use learnable embedding to encode text; w/o Query Decoder means that we directly use the average pooled feature from the encoder to predict a single referred output. We can see that the context encoder plays an important role in providing good textual representation for further multi-modal fusion. Query encoders are also quite important without which we also observe a big performance drop. For the ablation on query features, we observe that both context feature and phrase features are crucial without which the performance will decrease considerably. The table also showed that the network will not work without guidance of both context and phrase features since we cannot establish a correspondence between multiple queries and outputs in such a case.

In RES and REC tasks, the size of input image is a matter of trade off between performance and speed, which is largely effected by the network architecture. Despite that our model is designed to be able to process $640 \times 640$ images at real time speed, we also test our model at different input resolution for reference, as showed in Table 5. Note that for resolution 256 and 320, we adjust strides in the final stage of ResNet to keep the number of visual features sent into visual-lingual encoder roughly the same.

Table 6: **Comparison with Concurrent Work.** Performance on RefCOCO/RefCOCO+/RefCOCOg datasets [55]. Ours* denotes that pretraining is used. All methods use ResNet101 image backbone.

| Models | Visual Features | Pretrain Images | Multi-task | RefCOCO | | | RefCOCO+ | | | RefCOCOg | |
|--------|--------|--------|--------|------|-------|-------|------|-------|-------|-------|--------|
| | | | | val | testA | testB | val | testA | testB | val-u | test-u |
| *One-stage:* | | | | | | | | | | | |
| VGTR [10] | Bi-LSTM | None | × | 79.20 | 82.32 | 73.78 | 63.91 | 70.09 | 56.51 | 65.73 | 67.23 |
| TransVG [7] | BERT | None | × | 81.02 | 82.72 | 78.35 | 64.82 | 70.70 | 56.94 | 68.67 | 67.73 |
| Ours | BERT | None | ✓ | 82.23 | 85.59 | 76.57 | 71.58 | 75.96 | 62.16 | 69.41 | 69.40 |
| *Pretrained:* | | | | | | | | | | | |
| MDETR [23] | RoBERTa | 200k | ✓ | 86.75 | 89.58 | 81.41 | 79.52 | 84.09 | 70.62 | 81.64 | 80.89 |
| Ours* | BERT | 100k | ✓ | 85.65 | 88.73 | 81.16 | 77.55 | 82.26 | 68.99 | 79.25 | 80.01 |

## 4.6 Comparison with Contemporaneous Work

Concurrent and independent to us, very recently, there are some closely related works that use transformers for visual referring tasks [7, 10, 23]. We will briefly discuss some of the differences.

To begin, [7, 10] focus on REC, while our approach is formulated in multi-task setting and solves both REC and RES tasks simultaneously. In addition, our model is faster and is capable of grounding multiple contextualized phrases, while [7, 10] follow previous one-stage approaches and are only able to infer a single expression at a time; leading, in our case, to more accurate results. MDETR [23] leverages contrastive loss and soft token loss to help better match bounding boxes to phrases. In contrast, our method adopts a simple and straightforward method to pre-match bounding boxes to phrases using a one-to-one matching; this leads to simpler learning objective.

We provide quantitative comparison on REC task with these approaches, based on their reported numbers in Table 6. Compared to [7, 10], our model performs substantially better in all, with an exception of RefCOCO testB (where [7] is marginally better), datasets and splits. The biggest improvements can be seen on RefCOCO+, where our model is 10.4% better (or 6.76 points better), than the closest concurrent work of [7], on the Val split; similar sizable improvements are illustrated on other splits, *e.g.*, 9.2% on RefCOCO+ testB. In addition, our approach is considerably faster in runtime, since our model is able to handle multiple queries simultaneously (unlike [7, 10]). Compared to [23], in a pretrained model setting, we see that our model performs similarly on RefCOCO and marginally worse on RefCOCO+ and RefCOCOg. This difference can perhaps be attributed to two factors: (1) larger pretrain dataset and longer triaining schedule. (As reported in [23], MDETR takes 224 GPU days to pretrain, while the pretraining for our model is roughly 28 GPU days) (2) using more sophisticated language model (RoBERTa for [23] vs. BERT for us). Limited experiments in Supplemental Material show that indeed, the use of RoBERTa leads to certain improvements. In addition, we setup our method in multi-task setting to solve RES and REC task at the same time, so our formulation while perhaps marginally inferior on REC is more general overall.

## 5 Conclusions and Future Work

In this work, we present Referring Transformer, a one-step approach to referring expression comprehension (REC) and segmentation (RES). We jointly train our model for RES and REC tasks while enabling contextualized multi-expression references. Our models outperform state-of-the-art by a large margin on five / three datasets for REC / RES respectively, while achieving real-time runtime.

One limitation for our model is that we follow the setup in previous works [51, 52] and assume that each expression refers to only one region. In the future, we plan to explore learning to predict multiple regions for each referring entity if necessary. Large-scale multi-task pretraining has been demonstrated to be very effective for ViLBEERT and other similar architectures; this is complementary to our focus in this paper, and we expect such strategies to further improve the performance.

## 6 Acknowledgments and Disclosure of Funding

This work was funded, in part, by the Vector Institute for AI, Canada CIFAR AI Chair, NSERC CRC, NSERC Discovery and Discovery Accelerator Grants. Resources used in preparing this research were provided, in part, by the Province of Ontario, the Government of Canada through CIFAR, and companies sponsoring the Vector Institute www.vectorinstitute.ai/#partners. Additional hardware support was provided by John R. Evans Leaders Fund CFI grant and Compute Canada under the Resource Allocation Competition award.

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
