# Referring Transformer: A One-step Approach to Multi-task Visual Grounding
# Supplementary Material

**Muchen Li**[1,2]
muchenli@cs.ubc.ca

**Leonid Sigal**[1,2,3,4]
lsigal@cs.ubc.ca

[1]Department of Computer Science, University of British Columbia
[2]Vector Institute for AI     [3]CIFAR AI Chair     [4]NSERC CRC Chair

## 1   More Details for Referring Expression Segmentation (RES)

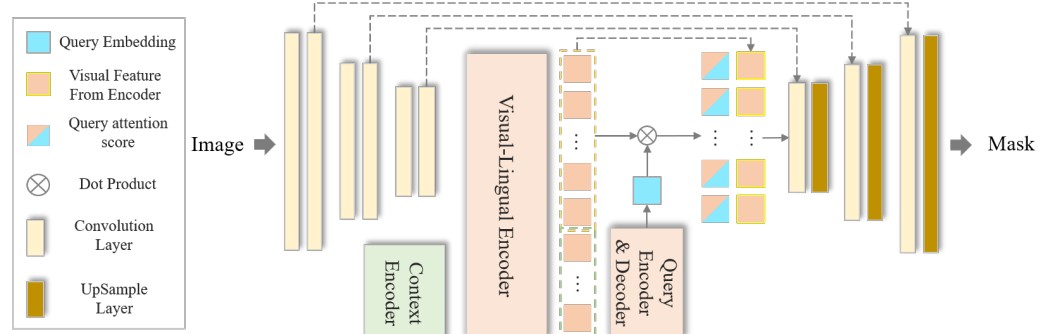

Figure 1: **RES Task Head.** A detailed illustration of our model for RES task.

We provide a more detailed illustration of our model for the RES task in Figure 1. With decoded query embedding from the query decoder and visual feature coming from visual-lingual decoder, a query attention score $\mathbf{S}_{att} \in \mathbb{R}^{M \times (HW)}$ is computed using a dot product. Here $M$ denotes the number of attention heads, which is $8$ in our implementation. The attention score is then concatenated with visual feature and sent into several up-sampling blocks (convolution layer with stride of 2). We also add residual connections from different stages of the ResNet backbone to help refine the up-sampled features. All convolution layers here use a kernel size of 3. The design is motivated by Mask R-CNN [3] and DETR [1].

## 2   Additional Implementation Details

**Pretraining.**     We use the description split of Visual Genome [6] for pretraining, it contains 100k images with an average of 40 region descriptions per-image. We pretrain our model with REC task on the Visual Genome dataset for 6 epochs. We set the learning rate at 1e-4 and decay it by 10x after 4 epochs. The trained model is then used to initialize the model for dataset-specific fine-tuning.

**ReferItGame / Flickr30k Training.**     On ReferItGame [5] and Flickr30k Entities [10], our model is trained for 90 and 60 epochs respectively, with learning rate decays on the 60th and 40th epoch. Following [1], we also use auxiliary loss to aid the training process.

**RefCOCO Training.**     For experiments on RefCOCO(+/g) [9, 11], since auxiliary loss is expensive for RES task, we first train our model with auxiliary loss on REC task for 60 epochs using a learning

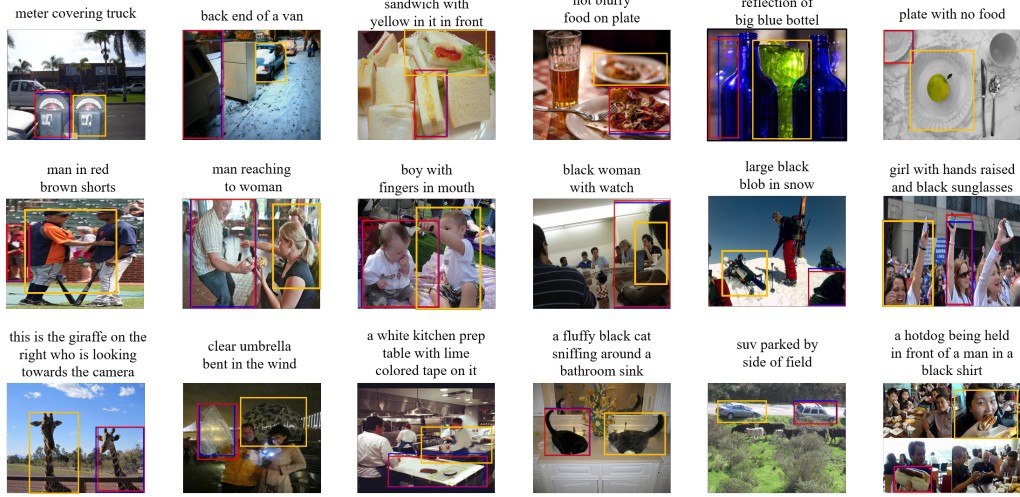

Figure 2: **Additional Qualitative Results on REC Task.** Orange, blue and red bounding boxes correspond to outputs from MCN [8], our model and the ground truth. The first row, second row and third row comes from RefCOCO+ testA, testB and RefCOCOg test set respectively.

rate of 1e-4. Then, we disable auxiliary loss and train the model jointly on RES and REC task for 30 epochs with learning rate of 1e-4 that decays on the 10th epoch.

# 3    Additional Results

**More Qualitative Comparison.**    Additional qualitative results on REC and RES tasks, compared to the previous multi-task framework of [8], are shown in Figure 2 and Figure 3 respectively. Note that the score of RES can benefit greatly from better localization of corresponding REC task. In Figure 3, to better compare the quality of generated referred masks, we compare the mask quality in the case where both MCN [8] and our method assuming correct REC localization.

**Ablation on Losses.**    We also conduct an ablation on the loss components. We can see that $L1$ loss is very important for REC tasks and Dice loss is most significant for RES. The gIoU and Focal loss terms also improve the performance, but by a much more modest margin. We further apply the Consistency Energy Minimization (CEM) proposed in MCN [8] to our model and observe consistent performance drop on RES task in Table 1. This shows that explicit constraint on multiple tasks will tend to degrade our model's performance.

Table 1: Ablations study on losses.

| Methods | RefCOCO REC | | |
| --- | --- | --- | --- |
| | val | testA | testB |
| RefTR | **81.82** | **85.33** | **76.31** |
| RefTR - $L1$ loss | 17.45 | 19.35 | 16.13 |
| RefTR - gIoU loss | 79.08 | 82.95 | 73.39 |
| RefTR + CEM loss [8] | 81.47 | 84.96 | 76.80 |
| Methods | RefCOCO RES | | |
| | val | testA | testB |
| RefTR | **69.94** | **72.80** | **66.13** |
| RefTR - Focal loss | 69.06 | 72.57 | 65.78 |
| RefTR - Dice loss | 64.04 | 67.09 | 60.03 |
| RefTR + CEM loss [8] | 69.17 | 72.17 | 65.33 |

**Results with Different Language Backbone.**    We also looked into the effect of different language backbone by replacing the BERT [2] model used in our model to RoBERTa [7] (also used by MDETR [4]). As we can see from Table 2, using RoBERTa gives us performance boost on most of the datasets. This shows that our model is scalebale and could potentially be benefited by larger backbone.

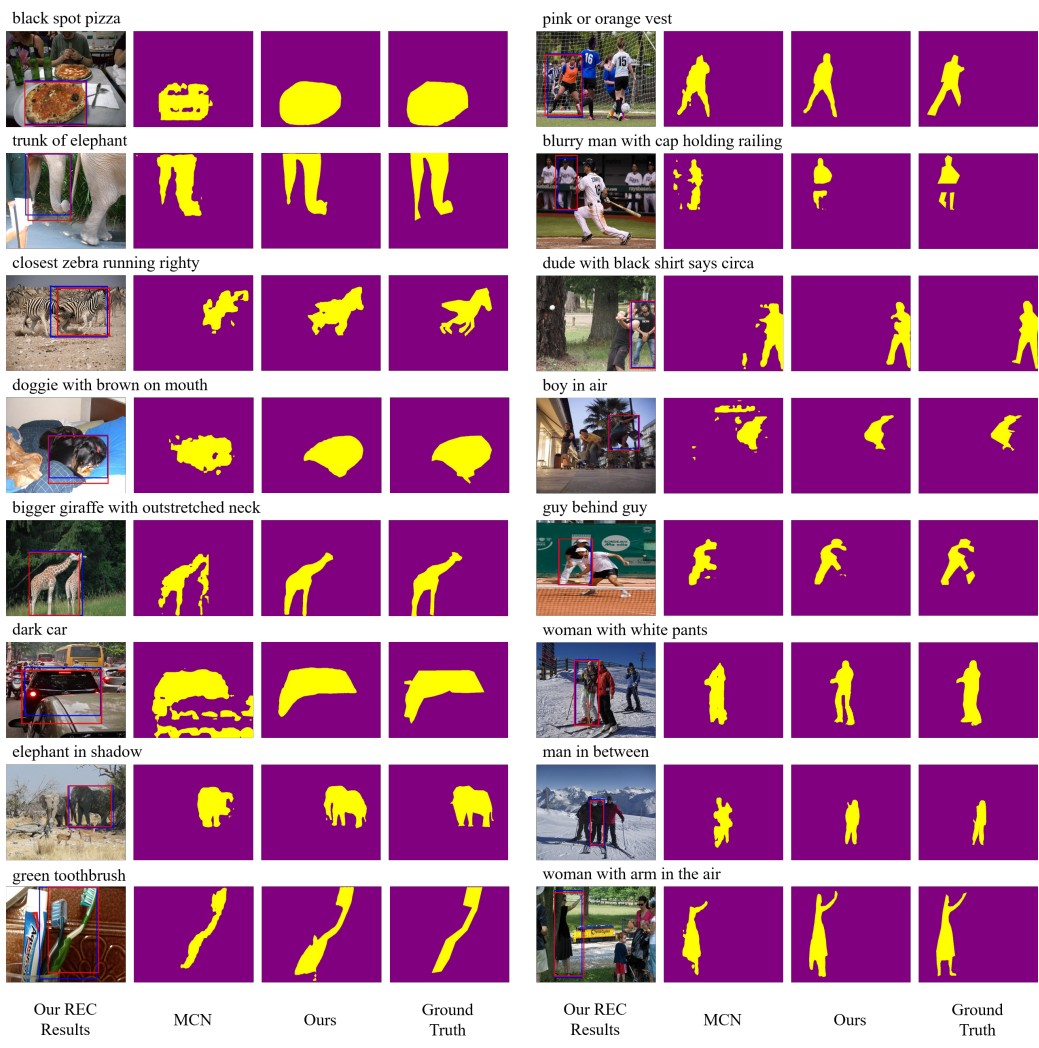

black spot pizza

pink or orange vest

trunk of elephant

blurry man with cap holding railing

closest zebra running righty

dude with black shirt says circa

doggie with brown on mouth

boy in air

bigger giraffe with outstretched neck

guy behind guy

dark car

woman with white pants

elephant in shadow

man in between

green toothbrush

woman with arm in the air

| Our REC Results | MCN | Ours | Ground Truth | Our REC Results | MCN | Ours | Ground Truth |

Figure 3: **Additional Results on RES Task.** Images come from RefCOCO+ testA and testB splits.

Table 2: Results on REC and RES tasks with RoBERTa.

| REC | RefCOCO | RefCOCO+ | RefCOCOg | Flickr30k |
|---|---|---|---|---|
| | val / testA / testB | val / testA / testB | val / test | test |
| RefTR | **81.82** / **85.33** / 76.31 | 71.13 / 75.58 / 61.91 | 69.32 / 69.10 | 78.13 |
| RefTR + RoBERTa | 81.52 / 85.01 / **76.90** | **71.41** / **76.82** / **61.95** | **69.40** / **69.78** | **78.56** |
| RES | RefCOCO | RefCOCO+ | RefCOCOg | - |
| | val / testA / testB | val / testA / testB | val / test | - |
| RefTR | **69.94** / **72.80** / 66.13 | **60.90** / 65.20 / **53.45** | 57.69 / 58.37 | - |
| RefTR + RoBERTa | 69.76 / **72.80** / **66.63** | 60.10 / **65.26** / 52.57 | **58.01** / **58.84** | - |

# 4  Asset License

For the assets that are used: 1. Huggingface uses a Apache License[1]. 2. RefCOCO(+/g) and ReferItGame also use a Apache License[2]. 3. Flickr30k Entities dataset use a license granted by Flickr term of use[3]. 4. Visual Genome use a Creative Commons Attribution 4.0 International License[4].

---

[1]https://github.com/huggingface/transformers/blob/master/LICENSE

[2]https://github.com/lichengunc/refer/blob/master/LICENSE

[3]https://www.flickr.com/help/terms/

[4]https://visualgenome.org/about