# OpenReview forum: "Referring Transformer: A One-step Approach to Multi-task Visual Grounding"
_NeurIPS.cc/2021/Conference — NeurIPS 2021 Poster_

### Official Review · Reviewer_zbBR · 2021-07-15

**Rating:** 6
**Confidence:** 5

**Summary:**

In this paper, the authors modify the DETR object detector to approach the referring expression comprehension and segmentation tasks (REC and RES). The language referring is taken as the decoder query to generate the box/mask prediction. The proposed method outperforms the SOTA on multiple standard REC and RES benchmarks.

**Limitations And Societal Impact:**

The limitations are discussed in the conclusion section.

**Main Review:**

Strengths:
1. The proposed method shows good accuracy and inference time on multiple REC and RES datasets.

2. The related work section provides comprehensive discussions of related areas (transformer detector, VLP, and REC/RES) as well as concurrent works.

Weaknesses:
1. Ablation on the claimed core contribution of decoder design (Line 56). The authors modify the decoder design in DETR to use the unique annotations in REC/RES, i.e., each language query corresponds to one and only one object box/mask. The experiment results verify that the proposed approach is one effective approach to modify DETR for REC/RES. However, it would make the claim stronger by comparing the proposed approach to alternative approaches, e.g., inputting the language query only in the encoder as in MDETR [23], and other possible decoder designs.

2. MCN [36] and this study both observe a large gain when jointly performing REC and RES. Thus, it might be helpful to provide the performance without multitasking as ablation, to help understand the gain from the proposed DETR-like architecture and multitasking.

3. Other than the modifications in the decoder, the proposed method somewhat lacks novelty (framework similar to DETR, multitasking studied by MCN [36]). Additional analyses on the source of improvements might make the paper stronger. Nonetheless, the proposed method serves as a new strong baseline for REC and RES.


A comment: as a future work, for RefCOCO and ReferIt datasets, it might be interesting jointly consider multiple queries on the same image. Such cross-sample relationships might further help the performance.


**Time Spent Reviewing:**

3

---

> ### Author Response · Authors · 2021-08-10
> **Respond to reviewer zbBR**
>
> We would like to thank the reviewer for valuable feedback and highlighting the strengths of our work. We address individual comments below.
>
>
> *1. It would make the claim stronger by comparing the proposed approach to alternative approaches, e.g., inputting the language query only in the encoder as in MDETR [23], and other possible decoder designs.*
>
> * As we briefly discussed in L128, MDETR[23] adopts a very different approach in the decoder. It follows DETR's detection pipeline to match the detected bounding boxes with phrases in the sentence. We tried to remove the query encoder and train the network using vanilla DETR with Hungarian matching; this results in a much lower performance with an accuracy of 66.86 on Flickr30k Test Split (vs 78.13 in our full model). MDETR also explores contrastive loss and soft token loss to help better match bounding boxes to tables, while our method pre-match bounding boxes to phrase positions using a one-to-one matching.
>
> * After the submission, we also spent some time aligning with MDETR's evaluation setup  (achieving an improved 79.09 performance for our model with ResNet101 and RoBERTAa backbone on Flickr30k test split). We will add more comparisons and analyses against MDETR in the revised version of the paper.
>
>
> *2. It might be helpful to provide the performance without multitasking as ablation.*
>
> * In Table 4 and L310 of the main paper, we provide ablations that measure the effect of multitasking. It is shown that multi-task training gives performance boosts on both RES (+ 1.11 in accuracy) and REC tasks (+ 2.69 mean IOU).
>
>
> *3.  Additional analyses on the source of improvements might make the paper stronger.*
>
> * Previous works on visual-linguistic tasks with transformers [5, 12, 33] adopt a two-staged process where visual features are extracted from proposal bounding boxes and ROI pooling. Our model, however, performs cross-modality alignment in a much earlier stage where grid features from the visual backbone are aligned with sentence features from lingual encoders by a cross-modal transformer. Such end-to-end design enables our model to attend to more fine-grained visual features with less semantic loss.
>
> * For the REC tasks, our model is different with respect to DETR in two key ways: (1) we get rid of Hungarrian matching in the loss and (2) we contextualize queries based on the aligned visual-linguistic features.
>
> * For the RES tasks, our model benefits from the precise localization of REC task (this has also been observed in [36]), as we implicitly restrict our model by letting it predict a joint representation and decode the attention map from the joint embedding. Finally, Dice loss also plays an important role for RES task. Please see loss ablations below (additional loss ablations and discussion can be found in response to Reviewer 3)
>
> | Methods               | RefCOCO Val/testA/testB(miou) |
> |  :----                | :----:                |
> | RefTR              | 69.94 / 72.80 / 66.13 |
> | RefTR - Focal loss  | 69.06 / 72.57 / 65.78 |
> | RefTR - Dice loss| 64.04 / 67.09 / 60.03 |

---

> > ### Comment · Reviewer_zbBR · 2021-08-26
> > **Post Rebuttal**
> >
> > Thank you for your responses.
> >
> > 1. I find both this paper's setting and the one in MDETR reasonable and interesting. The additional analyses are interesting and help readers to better understand the pros and cons when compared with MDETR.
> >
> > 2. The presented experiment addressed my previous concern on the REC+RES multitasking.
> >
> > 3. The mentioned two modifications over DETR seem straightforward adjustments to match the REC/RES setting, i.e., one query only matches one box. Nonetheless, I acknowledge the contribution of applying DETR to REC/RES.
> >
> > Overall, I would remain my initial positive rating and lean towards acceptance.

---

### Official Review · Reviewer_Ljjc · 2021-07-15

**Rating:** 6
**Confidence:** 3

**Summary:**

In this paper, the authors propose a one-stage multi-task framework for referring expression comprehension (REC) and segmentation (RES). They learn a single transformer based architecture with two separate heads for the bounding box regression and segmentation. It enables contextualized reasoning by taking as input the entire image, all referring query phrases of interest and the complete lingual context. They outperform state-of-the-art on various datasets for both these tasks and validate their design choices by ablation studies.


**Limitations And Societal Impact:**

The authors did not discuss the social impact of their work. It will be good to add some other limitations in the paper that the authors ran into when developing or training the models. One thing to consider during phrase grounding is the bias in the captions/phrases and the models producing errors when localizing men/women or commonly seen words, races or genders. Is it seen this commonly in the REC datasets as it is image captioning domains?


**Main Review:**

The authors design a general one-stage transformer architecture (inspired from DETR) for RES and REC tasks. Compared to previous approaches, their model is one-stage unlike the two-stage approaches (object detection followed by reasoning/grounding) and does not depend on manually tuned anchor definitions as used by previous one-stage approaches. The paper is well written, shows extensive experiments and is novel.

1. It is not very clear how this model architecture gets away without doing Hungarian matching as in DETR? Is it because each query corresponds to a single region in the image? As this is considered one of the advantages of the proposed model, it should be discussed more clearly.

2. In lines 141-142, does the query generator correspond to the query encoder? What is the output dimension/space from this query encoder? Is it (#ofphrases x feature dim)?

3. How many steps/iterations are done for the GCN in the query decoder? How does the model perform without updating using graphs and directly adding the cross attention layer?

4. How does this GCN work for RefCOCO dataset for instance, where the entire sentence is the phrase query?
Are the choice of losses in Eq 3 and 4 inspired from SOTA work in RES/REC or are they being used in such combinations in this paper? An ablation study on the combination of these lossed will provide a good idea about their usefulness.



**Time Spent Reviewing:**

3

---

> ### Author Response · Authors · 2021-08-10
> **Response to reviewer Ljjc**
>
> We would like to thank the reviewer for valuable feedback and acknowledging that our “paper is well written, shows extensive experiments and is novel.” We address individual comments below.
>
>
> *1. It is not very clear how this model architecture gets away without doing Hungarian matching as in DETR? Is it because each query corresponds to a single region in the image?.*
>
> * Yes, we followed the setting of previous one-stage methods [4, 51, 52] where each language query corresponds to one referred entity. The one-to-one mapping between referred expressions and regions allows us to remove the Hungarian matching.
>
>
> *2. In lines 141-142, does the query generator correspond to the query encoder? What is the output dimension/space from this query encoder?*
>
> * Thanks for pointing out the inconsistency of terminology here, we mean query encoder. The output dimension would be Batch size x Number of Phrases x Hidden dimension.
>
>
> *3. How many steps/iterations are done for the GCN in the query decoder? How does the model perform without updating using graphs and directly adding the cross attention layer? How does this GCN work for the RefCOCO dataset for instance, where the entire sentence is the phrase query?*
>
> * In our decoder, we do not use GCN in a recursive manner (with shared layer weights) as is done in Graph Convolutional Networks (the figure simply demonstrates that we have explicit interaction between queries in the same context). We stack 6 layers of the proposed model with independent weight. For the datasets where only one phrase is available, only the cross attention layer is needed, but we keep the original architecture for simplicity. We will revise the paper to clarify.
>
>
> *4.  An ablation study on the combination of these lossed will provide a good idea about their usefulness.*
>
> Here we provide an ablation of our proposed losses on the RefCOCO dataset.
>
> * For REC tasks, L1 loss and Giou loss are widely used in object detections. As we can see from the ablation, L1 loss plays a vital role, without which our model cannot converge; while Giou loss is also quite important in further boosting the performance on REC task.
>
> | Methods               | RefCOCO Val/testA/testB (acc) |
> |  :----                | :----:                |
> | RefTR             | 81.82 / 85.33 / 76.31 |
> | RefTR - L1 loss  | 17.45 / 19.35 / 16.13 |
> | RefTR - Giou loss| 79.08/ 82.95/ 73.39 |
>
> * For RES task, the focal loss is originally used in retinaNet for dense prediction; it is a weighted version of cross-entropy loss which can handle data imbalance. Dice loss helps us to focus more on segmentation, particular when there is imbalance in the number of background pixels and masked pixels. As we can see from the ablation table, while both losses contribute to the performance boost, dice loss plays a more pivotal role for RefCOCO dataset. We will add more analysis on losses in the revised version of the paper.
>
> | Methods               | RefCOCO Val/testA/testB(miou) |
> |  :----                | :----:                |
> | RefTR              | 69.94 / 72.80 / 66.13 |
> | RefTR - Focal loss  | 69.06 / 72.57 / 65.78 |
> | RefTR - Dice loss| 64.04 / 67.09 / 60.03 |
>
>
> *5. One thing to consider during phrase grounding is the bias in the captions/phrases and the models producing errors when localizing men/women or commonly seen words, races or genders. Is it seen this commonly in the REC datasets as it is image captioning domains?*
>
> * We do observe a little bias of our model which favours commonly seen words against rare ones, but we didn't observe severe bias that would have potential social impact. We will add analysis on the model's limitations in the revised paper.

---

> > ### Comment · Reviewer_Ljjc · 2021-08-25
> > **Post Rebuttal**
> >
> > I have carefully read the authors' response and they address my concerns. I appreciate the new experimental results provided by the authors and they give a better idea of the issue raised. I maintain my original rating.

---

### Official Review · Reviewer_aAnS · 2021-07-16

**Rating:** 6
**Confidence:** 3

**Summary:**

The paper proposes two modifications to DETR: 1) adding language input; 2) directly predict boxes from language phrase queries, instead of using random object queries and a set loss.

The result is a visual grounding model without limitation of the pre-trained object detection that is prevalent in previous vision-and-language models (e.g., ViLBERT/VL-BERT/UNITER…). It sets state-of-the-art results on grounding benchmarks such as Flirckr30K and RefCOCO. The model also opens door to building vision-and-language models without the limit of the pre-trained object detector.

Despite some limitations of the current model design, the paper stands in between traditional object detection and transitional vision-and-language modeling and shows promise and could lead to a good discussion.

**Ethics Review Area:**

["I don’t know"]

**Limitations And Societal Impact:**

Nothing I could think of.

**Main Review:**

Pros:

1. The model design is clever and makes sense. The paper, among contemporary work such as MDETR, is the first to show that it is possible to achieve good grounding performance without a pre-trained object detector.

2. In DETR, one interesting design is that they use object queries as anchor points. In this paper, the model design of removing the object queries and directly predicting boxes from the text (in a way, it is using text anchor points) is quite novel and inspiring for future work on grounding/detection.


Cons:
1. As noted in the paper, the model cannot ground one phrase to multiple boxes, which is a serious limitation.

2. Ideally, we would like a model where we simply input a sentence and an image, and the model could automatically ground entities in the sentence to the image. However, this seems not possible in the current model, as we have to specify the phrase span manually.

3. Pro 2 is not analyzed and discussed extensively in the paper. I will not count this as a reason for rejection but as it is one of the biggest design changes, it would be nice to have a detailed analysis.


Comments:
	A contemporary work MDETR takes a similar design (but with crucial differences). It could be beneficial to include it in the discussion (expectably against strength 2).

**Time Spent Reviewing:**

2

---

> ### Author Response · Authors · 2021-08-10
> **Response to reviewer aAnS**
>
> We would like to thank the reviewer for valuable feedback and acknowledging that our model design is “clever” and is “quite novel and inspiring for future work”. We address individual comments below.
>
>
> *1. As noted in the paper, the model cannot ground one phrase to multiple boxes, which is a serious limitation.*
>
> * We are mainly following the working paradigm of previous work on one-stage visual grounding [4, 51, 52] which all merge bounding boxes for multiple instances into one target box. This is sensible for most referring expressions in RefCOCO and other datasets, which tend to refer to a specific (object) instance. However, we admit this is a limitation which we plan to focus on in future works.
>
>
> *2. In the current model, we have to specify the phrase span manually.*
>
> * Referring expressions can be parsed from sentences ``automatically” using NLP tools. In fact, this is a common practice for some current and past datasets, e.g., ReferItGame [Kazemzadeh, EMNLP’14] used StanfordCoreNLP parser with noun sub-tree traversal for this. Other options include noun phrase extraction techniques based on constituency parsing. Since we are comparing to prior works on benchmark datasets, we simply use the phrases, and corresponding spans, provided by the datasets in our experiments.
>
>
> *3. Pro 2 is not analyzed and discussed extensively in the paper.*
>
> * We will add more detailed analysis and comparison with MEDTR and other contemporary work in the revised version of the paper (we did provide some very coarse high-level comparison in the Supplemental).
>
> * Meantime, we want to provide a preliminary discussion here. As we briefly discussed in L128, MDETR [23] adopts a very different approach in the decoder. MDETR follows DETR's detection pipeline to match the detected bounding boxes with locations in the sentence. We also tried to remove the query encoder and train the network using vanilla DETR with Hungarian Matching, this resulted in a much lower performance with an accuracy of 66.86 on Flickr30k Test Split (vs 78.13 in our full model). On top of this, MDETR [23] explores contrastive loss and soft token loss to help better match bounding boxes to phrases, while our method adopts a simple and straightforward method to pre-match bounding boxes to phrase positions using a one-to-one matching.
>
> * After the submission, we also spent some time aligning with MDETR's experimental setup for a more direct comparison (achieving an improved 79.09 performance for our model with ResNet101 and RoBERTAa backbone on Flickr30k test split). We will add more comparisons and analysis against MDETR in the revised version of the paper.

---

> > ### Comment · Reviewer_aAnS · 2021-08-26
> > **Response**
> >
> > Thank you for the rebuttal. For the point of "using text queries as anchors" v.s. MDETR's object query, MDETR's design would allow multiple boxes mapped to one text query while the design in this paper also has its own merits. Thus, I think it would be good as long as the paper points out the difference and makes a discussion. I would not hold it against the paper that the absolute numbers are lower than MDETR.

---

> > > ### Author Response · Authors · 2021-08-26
> > > **Respond to reviewer aAnS**
> > >
> > > We want to thank you again for your valuable feedback! Yes, MDETR's design allows it to ground to multiple regions while our paper focus on the one-to-one mapping between grounded regions and text queries, we will make this clear in the revised paper. At the same time, we are aligning our scores with MDETR by replacing BERT with Roberta used in MDETR. More results and discussions will be added to the revised paper.

---

### Official Review · Reviewer_GFR8 · 2021-07-17

**Rating:** 6
**Confidence:** 3

**Summary:**

This paper concerns referring expression comprehension (REC) and segmentation (RES). It proposes a unified one-step approach for both tasks through multi-task training. The proposed architecture is based on DETR, with a twist on context encoder to encode query caption and query encoder to encode each referring expression / entity. Comprehensive experiments are conducted over mainstream REC and RES datasets. Significant empirical improvements are observed.

**Limitations And Societal Impact:**

Appear to be.

**Main Review:**

The proposed idea is simple and effective. The model design is technically sound and is motivated clearly. Implementation details are handled elegantly, for instance, the unified context/query input format when given phrases vs. captions. The experimental improvements are evident across the board. Ablation studies on multi-task training and each component of the network are informative. Overall, the work is performed with great quality and pushes the boundary of visual grounding.

**Time Spent Reviewing:**

2

---

> ### Author Response · Authors · 2021-08-10
> **Respond to reviewer GFR8**
>
> We would like to thank the reviewer for their efforts and a very positive review.

---

### Decision · Program_Chairs · 2021-09-28

**Decision:**

Accept (Poster)

**Comment:**

The work enables one-stage, end-to-end visual grounding using Transformer. The architecture is designed based on DETR, by replacing the object queries with phrase encoding, and equipped with several additional components for the grounding task. Overall, the reviewers unanimously agree the work makes a nice contribution with a simple and effective architecture, which cleverly extends the DETR architecture that was designed for object detection. The reviewers also found the experimentation sufficient, where the work showed good accuracy on a number of datasets. The authors' rebuttal clarified a few reviewer concerns, which further strengthens their confidence for accepting the paper.

**Consistency Experiment:**

NeurIPS has a long history of experimentation. In 2014, NeurIPS ran an experiment in which 10% of submissions were reviewed by two independent committees to quantify the randomness in the review process. This year, we repeated a variant of this experiment to see how the quality of the review process has changed over time.  This paper was part of the experiment and was therefore assigned to two committees (consisting of reviewers, an Area Chair, and a Senior Area Chair) that reached independent decisions.  If both committees made the same recommendation, this recommendation was followed. If a single committee recommended acceptance, the paper was accepted (with the exception of a few cases in which the other committee identified what we considered a fatal flaw, e.g., an error in a key result).

This copy’s committee reached the following decision: **Accept (Poster)**

The other committee assigned to the paper recommended **Reject**.  You can find the other set of reviews, along with any follow up discussion with the authors here:
https://openreview.net/forum?id=j7u7cJDBo8p